# Biological Cell Investigation of Structured Nitinol Surfaces for the Functionalization of Implants

**DOI:** 10.3390/ma13153264

**Published:** 2020-07-23

**Authors:** Isabell Hamann, Ute Hempel, Christian Rotsch, Mario Leimert

**Affiliations:** 1Department of Medical Engineering, Fraunhofer Institute for Machine Tools and Forming Technology, Dresden, 01187 Saxony, Germany; christian.rotsch@iwu.fraunhofer.de; 2Department of Spine Center, Asklepios Orthopädische Klinik Hohwald, Neustadt i. Sa., 01844 Saxony, Germany; 3Institute of Physiological Chemistry, Medical Faculty Carl Gustav Carus TU Dresden, Dresden, 01307 Saxony, Germany; hempel-u@mail.zih.tu-dresden.de; 4Department of Neurosurgery and Spine Surgery, Sächsische Schweiz Kliniken GmbH, Sebnitz, 01855 Saxony, Germany; m.leimert@asklepios.com

**Keywords:** additive structuring, biocompatibility, human bone marrow stromal cells, implant surface, laser structuring, primary implant stability increase, shape memory alloy anchor

## Abstract

Expandable implants including shape memory alloy (SMA) elements have great potential to minimize the risk of implant loosening and to increase the primary stability of bone anchoring. Surface structuring of such elements may further improve these properties and support osteointegration and bone healing. In this given study, SMA sheets were processed by deploying additive and removal manufacturing technologies for 3D-printed surgical implants. The additive technology was realized by applying a new laser beam melting technology to print titanium structures on the SMA sheets. The removal step was realized as a standard process with an ultrashort-pulse laser. The morphology, metabolic activity, and mineralization patterns of human bone marrow stromal cells were examined to evaluate the biocompatibility of the new surface structures. It was shown that both surface structures support cell adhesion and the formation of a cytoskeleton. The examination of the metabolic activity of the marrow stromal cells on the samples showed that the number of cells on the laser-structured samples was lower when compared to the 3D-printed ones. The calcium phosphate accumulation, which was used to examine the mineralization of marrow stromal cells, was higher in the laser-structured samples than in the 3D-printed ones. These results indicate that the additive- and laser-structured SAM sheets seem biocompatible and that the macrostructure surface and manufacturing technology may have positive influences on the behavior of the bone formation. The use of the new additive technique and the resulting macrostructures seems to be a promising approach to combine increased anchorage stability with simultaneously enhanced osteointegration.

## 1. Introduction

New concepts for advanced implant designs and geometry modifications aim to enhance bone anchorage to achieve subsequent stability, and are currently within the research focus of a number of groups [1,2,3,4,5,6]. Previous studies by our own group were concerned with the development of expandable screws [7]. Additionally, another preliminary study by our own group showed that unprocessed SMA sheets are biocompatible and appropriate for application in orthopedic surgery [8].

It has been shown that the use of shape memory alloy (SMA) made of nickel-titanium has great potential to increase primary stability in bone [9,10,11]. Additionally, elements or surface patterns of surgical implants combined with SMA may enhance bone anchoring stability, thus improving the functionality and durability of the implant when loaded under physiological stress. Improvements in the implant design seem necessary to allow for optimal surgical treatment of the elderly, considering the ongoing demographic change in industrialized nations. Osteoporosis is one of the surgical challenges, as the primary anchorage stability of standard implants is less likely to be maintained, requiring adjustments in the implant design.

SMA expansion sheets developed as part of a previous study [7] were designed to expand in cancellous bone similarly to a dowel, increasing the contact surface and wedging within the bone [7]. These previously designed sheets had smooth surfaces, which resulted in increased anchoring stability. However, the smooth surfaces of the actuator sheets still need to be improved. 

The aim of this study was to evaluate the biocompatibility of structured SMA sheets manufactured in a two-step additional manufacturing and material removal process (Figure 1). Using a human bone marrow stroma cell model (hBMSC), the effects of chemical alterations introduced by this new manufacturing process were studied. It was hypothesized that that the surface treatments would have no negative effects on the biological cell investigation or biocompatibility.

## 2. Materials and Methods 

### 2.1. Refinement

#### 2.1.1. Basic Material of the Actuator Sheets

Thirty-six rectangular plates with a base of 36 mm^2^ (approximately 6 × 6 mm) were separated from a 1-mm-thick Nitinol sheet (Memry, Bethel, CT, USA; alloy S_1p000_0115_B, further specifications in Table 1) using a water jet cutting machine (5-axis fine blasting machine, custom design by the TU Chemnitz, Chemnitz, Germany).

#### 2.1.2. Structuring Process of the Actuator Plates

Two different established technologies were used to structure the SMA sheet surfaces, namely three-dimensional (3D) printing and laser-based removal. In each case, 18 SMA sheets were processed.

Three-dimensional printing: For this additive structuring method, a new application of the laser beam melting technology was deployed. Therefore, untreated Ti6Al4V (VDI 3405-2.4 grade 5) was applied to 16 SMA sheets (Concept Laser M2 Cusing, Coburg, Germany). The specifications were as follows: laser power = 105 W, scanning velocity = 600 mm/s, layer thickness of the pyramid-shaped structures = 25 μm. For better adhesion of the titanium to the SMA sheets, these were exposed and sandblasted once prior to the construction job. The geometry of the pyramid shapes generated here was as follows: 0.5 × 0.5 mm, height 0.9 mm (Figure 2).

Laser cutting: The remaining 16 samples were structured using a standard removal technology with an ultrashort-pulse laser (Spirit 1030-100 SHG, Spectra-Physics MKS, Andover, MA, USA). The pyramid structures were lasered with a 40 W laser and a 4 MHz pulse fetch frequency, resulting in a pulse energy of 10 μJ. The focus diameter averaged 30 μm. The pyramid structure arose from crossed lines, forming a grid. Every crossed line was formed by a number of individual parallel lines at a distance of 5 μm, repeated 25 times. The V-shaped trenches were realized by reducing the number of parallel lines and by piecewise removal. This piecewise removal was conducted due to the low erosion rates of the USP laser to weaken the thermal input. The pyramid shapes had the following geometries: 0.3 × 0.3 mm base, 0.5 mm in height (Figure 3). 

### 2.2. Characterization of the Actuator Plates

The experiments were performed with hBMSCs from three independent biological donors. To characterize the given structured surfaces in vitro, cell experiments with primary human bone marrow stromal cells (hBMSCs) were performed. The hBMSC morphology was examined with immunofluorescence staining. The metabolic activity of hBMSCs, namely the activity of cellular (mitochondrial) dehydrogenases, was determined with the MTS assay to evaluate the biological suitability of the structure and its biocompatibility. As a third parameter, cell-associated calcium phosphate deposition was measured to determine a late stage of osteogenic differentiation.

#### 2.2.1. Sample Preparation

To sterilize and remove all contamination, the SMA samples were cleaned twice with 70 mass% ethanol for 10–15 min each and air-dried overnight.

#### 2.2.2. Medium

The basal medium (BM) consisted of Dulbecco’s modified Eagle’s medium (DMEM) with 10% heat-inactivated fetal bovine serum (HI-FCS) and penicillin–streptomycin (Pen/Strep). For the differentiation medium (OM/D), the BM was supplemented with 10 mM β-glycerophosphate, 10 nM dexamethasone, and 300 μM ascorbate.

#### 2.2.3. Seeding and Evaluation

First, 10,000 hBMSCs were seeded in 50 μL of BM directly onto the surfaces of the samples (placed in a 24-well cell culture plate. After 2 h in the incubator (humidified atmosphere, 5% CO_2_) at 37 °C, BM was filled up to 1 ml to cover the samples with medium. The medium was changed twice weekly and OM/D was used from day 4 after seeding. Prior to the analyses, the samples were placed in new wells.

To evaluate the hBMSC behavior when exposed to the SMA sheets, cell morphology was assessed 24 h after the seeding by immunofluorescence staining of the cytoskeleton and the cell nuclei. F-actin and vinculin were examined. The metabolic activity was determined at days 2 and 4 after seeding with an MTS assay 3-(4,5-dimethylthiazol-2-yl)-5-(3-carboxymethoxyphenyl)-2-(4-sulfophenyl)-2H-tetrazolium-Test parameters for the determination of metabolic activity). To quantify the mineralization, the deposition of calcium phosphate was determined at day 22 following the hBMSC seeding.

#### 2.2.4. Statistical Analysis

The experiments were performed with hBMSCs from three independent biological donors. The differentiation potential of the donors was checked before in a separate experiment. Each cell preparation was used in duplicate (two independent samples) and each measurement was done in triplicate. GraphPad Prism 8.4.3 software (San Diego, CA, USA) was used for statistical analyses. For the MTS assay, statistical significance was analyzed by two-way analysis of variance (ANOVA)with Bonferroni’s post-test. To analyze the calcium phosphate results, an unpaired *t*-test was applied.

## 3. Results

### 3.1. Morphology

By examining the cell morphology with immunofluorescence staining, it could be proven whether hBMSCs adhered and spread on the surfaces, thus changing their spherical shape to become planar within 24 h following the seeding. As seen in Figure 4, cell nuclei (blue fluorescence) and the cytoskeletal proteins F-actin (green fluorescence) and vinculin (purple fluorescence) are present.

The co-localization of both F-actin and vinculin appears as white spots. In both surface structures, the cells adhered in the pyramidal valleys, expressing cytoskeletal proteins. Optimal and sharp imaging was not possible because of the three-dimensionality of the samples.

### 3.2. Metabolic Activity

Metabolic activity was determined at days 2 and 4 following the seeding with the MTS assay. It was shown that hBMSC increased significantly on the 3D-printed samples, whereas on the laser-structured samples the metabolic activity decreased slightly but insignificantly. As compared to the laser-structured samples, more cells were counted on the 3D-printed samples at day 4 (Figure 5).

### 3.3. Mineralization

The ratio of calcium to phosphate was close to the theoretical value of 1.67, which is characteristic for bone mineral hydroxyapatite. On the 3D-printed samples, the values were significantly lower.

On the laser-structured samples, 0.85 ± 0.54 μmol calcium and 0.88 ± 0.32 μmol phosphate were determined, on the 3D structured samples 0.08 ± 0.01 μmol calcium and 0.26 ± 0.03 μmol phosphate were determined (Figure 6). Cell-free samples, which were cultured under the same conditions, were used as the control and analyzed for calcium phosphate accumulation.

## 4. Discussion

Surface structuring seems to be a key factor in contemporary implant technology to improve bone anchoring. The main focus is set on the microstructuring of surfaces, which improves cell growth on the implant surface [12,13,14,15,16]. Secondary stability, however, is critically dependent on osteoblasts successfully adhering to the implant. On the other hand, chemical or structural surface treatments may also be used to reduce contamination with bacteria to prevent septic loosening [13,17]. Any effective secondary stability is based on a successful primary stability. This provides information about the anchor of the implant directly after implantation. Due to demographic changes and the resulting increased supply of osteoporotic bone, the primary stability of implants cannot always be guaranteed [18,19,20,21]. 

Our own preliminary tests with new cementless implant systems, therefore, involved additional sheets made of SMA [7].

The first successful tests gave evidence in favor of enhanced primary stability. Two different processing technologies were applied to increase the primary stability and functionalization of these sheets. Both processes are already used in medical technology.

### 4.1. Structure Technology

To increase the surface and anchorage stability, the sheets were additionally treated with two different methods for structuring. Both additive and removal techniques were subsequently deployed. These two methods were previously established as standard procedures in the manufacturing of medical devices such as stents and joint implants [11,22,23] For the additive manufacturing step, a standard laser beam melting process for individualized medical implants was used [24], supplemented with a new process for printing structures on SMA sheets. The removal treatment was a standard technique, which helped structure and section large-area SMA sheets [25].

The new additive laser beam melting (LBM) technique introduced here to refine SMA-based advances in surface structuring was achieved by applying titanium as an additional material. LBM is a technology that is classified as a beam melting process. The advantage of LBM is the free shaping, allowing for rapidly discontinued tools or formed components. In the medical device industry, 3D printing is presently used primarily to manufacture patient-specific implants [11]. The capacity of LBM as a 3D printing technology makes it feasible to introduce refined surface roughness as an appealing alternative to more technically challenging methods involving more steps in manufacturing. The application of 3D printing on a metal sheet is a promising new development, which was carried out in the context of this given investigation for the first time. When applying the titanium layers, a laser is used to scan the pre-programmed structure and to melt the titanium powder particles situated on top of the sheet. After solidification, the titanium forms a solid material layer. During this process, a high thermal input is generated for a short time, which is also transferred to the SMA sheet. Due to the heat, structural changes occur in the SMA, damaging the superficial oxide layer and allowing toxic nickel ions to release. To date, no comparable studies are available on the biocompatibility and cell behavior of printing SMA sheets treated this way. 

On the other hand, the laser structuring of the sheets occurs at a lower temperature. The material processing of SMA parts is an integral part of stent manufacturing [22,23] 

Full surface covering and structuring of SMA sheets using laser processing has not yet been biologically researched. The process is an erosive processing technology. The material is removed by short (femto- to picoseconds) high-energy laser pulses. This allows for micromachining (cutting, drilling, ablation, or structuring) of the smallest structures, ranging up to entire surfaces. The laser works very precisely and the process is free of burrs, furrows, or material residues, which is important for medical device technology to ensure a hygienically “unpolluted” surface. In addition, consistent quality can be ensured when creating structures because the laser has no wear elements. The critical aspect in laser processing is the opening of the oxide layer when removing the material. This exposes the natural oxide layer, meaning nickel ions can be released. However, the most important advantage when processing SMA sheets is the low heat influence, meaning that structural changes and alterations to the thermal or superelastic properties of the SMA can be excluded [26,27,28]. Due to the small size of the laser focus, it has the capacity to realize minute structures and to adjust to the optimal surface roughness for cell ingrowth. In contrast to new 3D printing techniques, it is currently impossible to reduce the size of the structures further, as the resulting surface adhesion between the base of the titanium pyramids and the SMA sheet would be too small. However, this technology opens up the opportunity for future research approaches that may potentially also allow printing of SMA material onto SMA sheets [29,30]. This seems to have great potential due to the enhanced surface adhesion between materials with identical or similar properties.

The biocompatibility of the new SMA design was assessed using an hBMSC model. The biocompatibility assessment was important, since the new two-step manufacturing introduced here may cause alterations in the chemical composition of SMA sheets. SMA forms a natural titanium oxide layer (TiO_2_), similarly to titanium alloys. The TiO_2_ layer helps protect the underlying metal from corrosion and removes toxic nickel ions from the implant surface, thereby acting as a physical barrier to nickel oxidation and protecting the SMA sheets from corrosion [31,32,33] and subsequent failure. In contrast, damage to the surface facilitates particle release of nickel ions. Consequently, it was important to examine the surfaces of SMA sheets using an established in vitro protocol with hBMSCs to assess biocompatibility and ingrowth.

### 4.2. Morphology

Owing to the rough 3D topography of the nitinol samples, it was not possible to obtain sharp micrographs, even with a confocal laser-scanning microscope. Therefore, an exact assessment of their orientation on the surface was not possible due to the three-dimensionality of the samples. Nevertheless, the immunofluorescence images demonstrate that the cells adhered on the samples (nuclei are present) and were able to express cytoskeletal proteins, such as fibrillary actin fibers and vinculin. It was seen that the cells adhered and spread out in the pyramid grooves. Both surface structures of the pyramids supported cell adhesion and the formation of a cytoskeleton. The main difference was the size of the pyramid valleys, where more cells could attach and spread over a larger area. It was shown that a natural titanium passivation still formed following the processing. However, the danger of layer damage or nickel ion release is higher in laser-structured sheets when compared to 3D printed materials. In this connection, surfaces are removed over a large area, whereas the 3D prints may form a biocompatible surface layer. This additional layer may help prevent nickel release following friction between the implant and the bone, especially under dynamic conditions.

### 4.3. Metabolic Activity

Examination of the metabolic activity by the MTS assay showed that cells proliferated on both surfaces and that the cell count increased. Due to the proof that the cells proliferated, it can be assumed that the surface is not cytotoxic. For this reason, an exact cytotoxic detection test, which determines the integrity of the cell membrane, was not used. In addition, previous studies by our group investigated the metabolic activity of nickel titanium (SMA material), titanium (Ti6Al4V), and coated nickel titanium materials. In comparison, it was seen that the untreated nickel titanium samples in the previous study showed a comparatively high metabolic activity similar to the structured 3D-printed sample prepared in this study. The laser-structured sample showed a lower but insignificant metabolic activity [8]. Therefore, it can be assumed that cell proliferation is not affected by the processing.

In detail, the examination of the metabolic activity of the hBMSCs on the samples showed that the number of cells on the laser-structured samples was lower when compared to the 3D-printed ones. Due to the processing and opening of the oxide layer, it may not have completely reformed, thus inhibiting cell growth. The primary aim of the treated surfaces is to enhance cell growth; however, other factors exist and can have negative impacts on cell growth, e.g., bacteria [13,17]. Laser-structured surfaces may have the potential to enhance osteoblast differentiation and growth, reduce bacteria adhesion, and at the same time offer high anchorage stability compared to unstructured materials. Further investigations regarding cell growth, reduction of bacterial adhesion, and the influences of structuring and manufacturing technology are needed to determine the combined functionalization. On the other hand, it was seen that the 3D printing technology may have potential as a standard structuring process, and therefore may be considered an alternative technique. An additional layer may also serve as a protective layer, which would particularly suitable for structuring using individualized design.

### 4.4. Mineralization

A characteristic feature of osteoblasts is the accumulation of calcium phosphate. Non-specific alkaline phosphate tissue activity is often used as an early differentiation marker, because the enzyme provides the phosphate amount necessary for proper calcium phosphate precipitation.

The calcium phosphate deposition correlates with the non-specific alkaline phosphatase tissue activity. Here, we used mineralization to characterize the late osteogenic differentiation status of the cells [34]. It was seen in this given study that calcium phosphate accumulation was higher in the laser-structured samples than in the 3D-printed ones. Here, the surface structure and production technology had an influence on the mineralization patterns. Thus, laser-structured surfaces may potentially have a positive effect as anchoring structures, resulting in accelerated formation and growth of osteoblasts. The in vitro investigations with hBMSCs showed that the materials were biocompatible and that the surface structure and manufacturing technology influenced the cell behavior. However, the samples were less suitable for microscopic examinations. The biochemical parameters and the quantification of the cell proliferation, overall count, and osteogenic differentiation were well determined. Although the laser-structured samples showed characteristics in favor of facilitating cell growth at a microstructure surface roughness of 10–100 mm, as described in the literature [12,35,36], the 3D-printed samples yielded better results. This also demonstrated that the new 3D printing technology might have the potential to be applied in future to structure SMA sheet surfaces. The flexible design process of the anchor formed on the SMA sheets and the simultaneous tuning of the cell adhesion and growth seem promising. Another advantage is the usage of thinner SMA sheets due to the extra additive layer applied for structuring. This may help reduce the costs of the expensive SMA materials and to reduce toxic nickel ion release. 

In further investigations, different structural features should be studied, including the size, geometry, and surface roughness of the anchor elements. Previous investigations have mainly been concerned with nano- and microstructured surfaces [37,38,39,40]. The macrostructures (adapted to bone density) shown here use a novel approach to combine increased anchorage stability with simultaneous osteointegration.

## 5. Conclusions

This study has shown the potential of the application new technology (3D printing on SMA sheets and laser structuring) for the structuring of SMA surfaces. It was shown that these technologies are biocompatible and that the cells are formed on the respective surfaces. Further studies should involve the macroscopic structures and their influence on cell growth to differentiate the relationship between the macrostructure and cell growth.

## 6. Research Involving Cell Lines

Human bone marrow stromal cells (hBMSCs) were isolated from bone marrow aspirates collected from healthy donors at the Dresden Bone Marrow Transplantation Center of the Carl Gustav Carus University Hospital (Dresden, Germany). The study was approved by the local ethics committee (EK263122004, EV466112016). The hBMSCs were isolated according to Oswald et al. 2004 [41]. For each experiment, the hBMSC preparations of three individual donors were not pooled or used in passages 3 or 4 [41].

## Figures and Tables

**Figure 1 materials-13-03264-f001:**
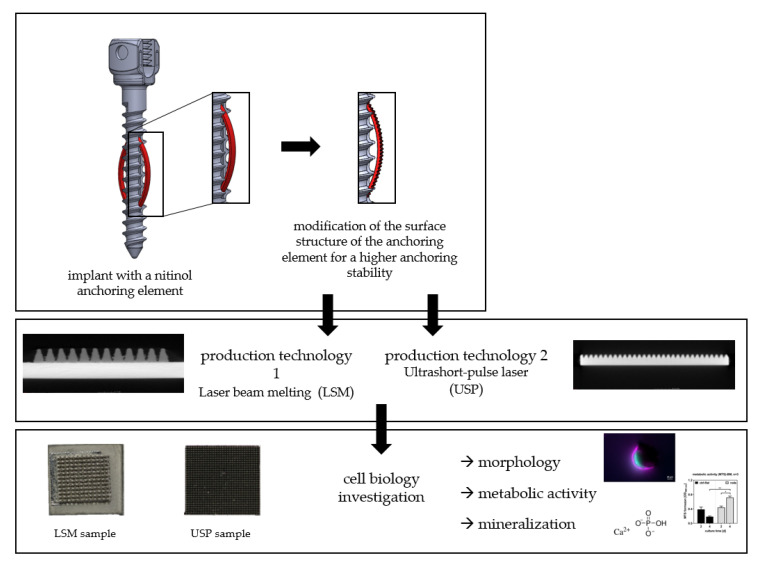
New screw design with nitinol sheets.

**Figure 2 materials-13-03264-f002:**
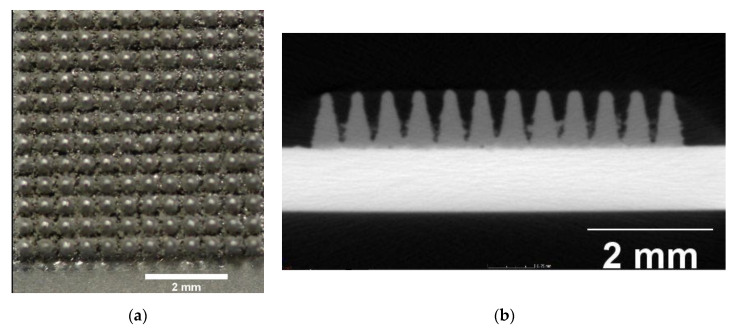
(**a**) 3D-printed-structured surface (top view); (**b**) Laser-structured surface microcomputed tomography (μCT) image: laser-structured pyramids on super elastic sheet metal; basic area: (a × b) = 0.2 × 0.2 mm, height (z) = 0.2 mm.

**Figure 3 materials-13-03264-f003:**
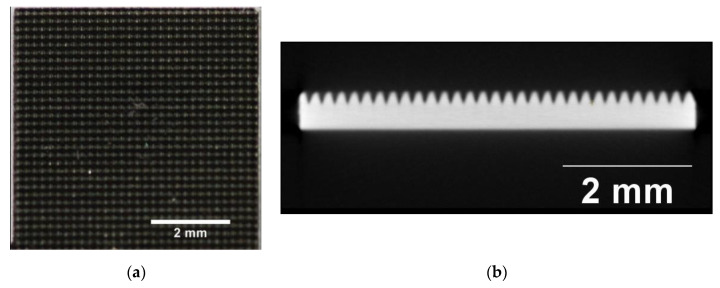
(**a**) Laser-structured surface (top view); (**b**) Laser-structured surface microcomputed tomography (μCT) image: laser-structured pyramids on super elastic sheet metal; basic area: (a × b) = 0.2 × 0.2 mm, height (z) = 0.2 mm.

**Figure 4 materials-13-03264-f004:**
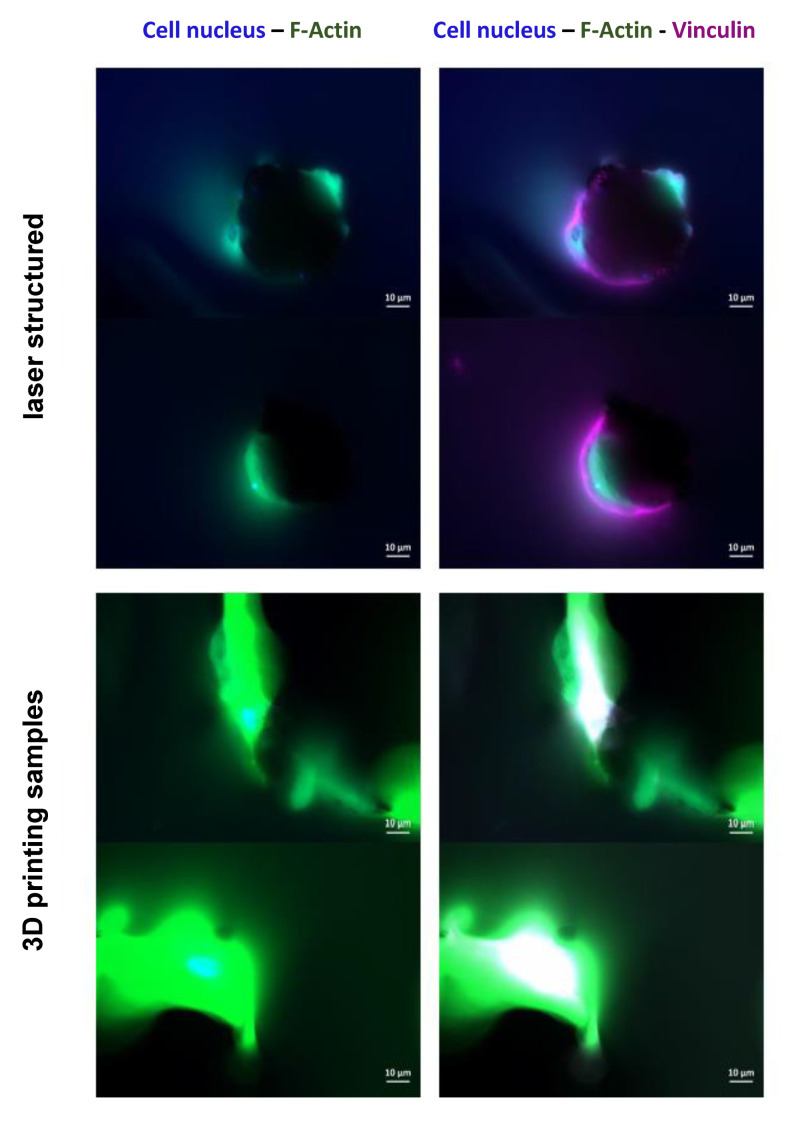
Immunofluorescence staining of human bone marrow stroma cells (hBMSCs) 24 h after sowing. Cell nuclei appear in blue (DAPI staining). Due to the three-dimensionality of the surfaces, the images lack clear sharpness and provide only a representation of cellular structures. Fibrillary actin (F-actin; green fluorescence)- and vinculin (violet fluorescence)-positive cells are shown.

**Figure 5 materials-13-03264-f005:**
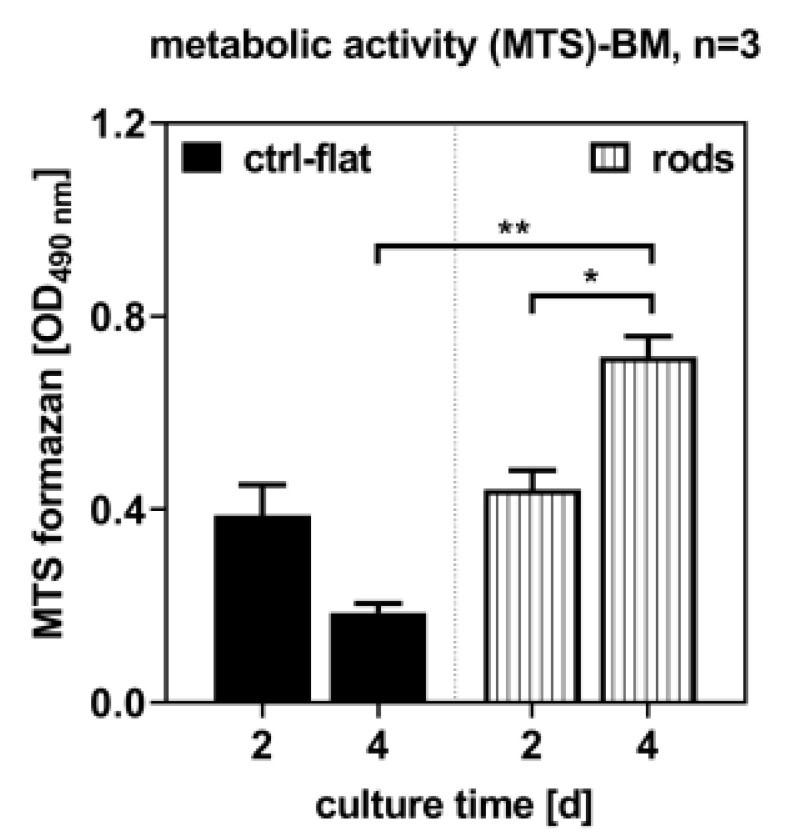
Metabolic activity and number of living cells at days 2 and 4 following the sowing of the cells. Significant differences indicated with * (*p* < 0.05) and ** (*p* < 0.01), n = 3; (BM = basal medium).

**Figure 6 materials-13-03264-f006:**
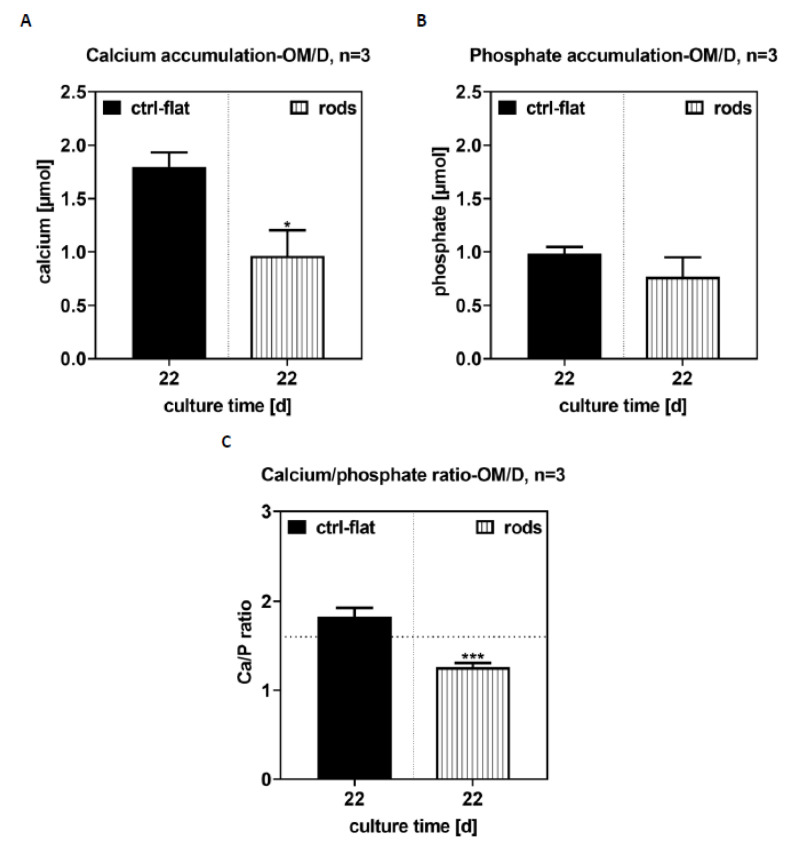
Quantification of calcium (**A**) and phosphate (**B**) on day 22, showing the ratio of Ca to phosphate (**C**). Significant differences indicated with * (*p* < 0.05) and *** (*p* < 0.001), n = 3.

**Table 1 materials-13-03264-t001:** Material properties.

Material density	6.45 kg × m^−3^
Coefficient of thermal expansion (martensite)	6.6
Coefficient of thermal expansion (austenite)	11
Specific electrical resistance (martensite)	76 Ω·mm^2^ × m^−1^
Specific electrical resistance (austenite)	82 Ω·mm^2^ × m^−1^

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
