# Peer review of "Biological Cell Investigation of Structured Nitinol Surfaces for the Functionalization of Implants"

_materials, 2020, doi:10.3390/ma13153264_

Round 1

Reviewer 1 Report

The aim of the study “ Cell biological investigation of structured nitinol surfaces for the functionalization of implants “ is to evaluate biocompatibility of structured SMA sheets manufactured in a two-step additional manufacturing and material removal process.

The data that the authors obtain are very interesting since that it seem  the new treatments of the surface  may further improve and support osseointegration and bone healing.

The article is accurately structured and the methodology is appropriate although it is necessary improve it.

Minor Changes

In the material and methods section,

In general the cell biological investigation it is necessary improve; the authors are not very precise in describing the methodology, making it impossible for other authors to replicate it.

In relation to Statistical Analysis, the authors do not mention it is the work; and it is important to know what methods they have used

Reviewer 2 Report

The manuscript entitled: "Cell biological investigation of structured nitinol surfaces for the functionalization of implants" presents biological evaluation of implants. This study includes assessment of morphology, metabolic activity, and mineralization of hBMSC. 

This manuscript may be promising for publication in Materials, but some remarks must be considered.

1) In my opinion the number of cell culture experiments is too low. Did the authors evaluate cytotoxicity of implants? I want also see the results from cell proliferation. Please, evaluate the characteristic markers of cell differentation into osteoblasts.

2) The quality of Figure 4 is not acceptable. Please, make new photos.

Round 2

Reviewer 2 Report

I accept this manuscript in present form.